# Efficacy and Complication Rates of Percutaneous Vertebroplasty and Kyphoplasty in the Treatment of Vertebral Compression Fractures: A Retrospective Analysis of 280 Patients

**DOI:** 10.3390/jcm13051495

**Published:** 2024-03-05

**Authors:** Jan Cerny, Jan Soukup, Kadzhik Petrosian, Lucie Loukotova, Tomas Novotny

**Affiliations:** 1Department of Orthopaedics, Faculty of Health Studies, Jan Evangelista Purkyne University in Usti nad Labem and Masaryk Hospital, 401 13 Usti nad Labem, Czech Republic; jan.cerny2@kzcr.eu (J.C.); soukup07@kzcr.eu (J.S.); kadzhik.petrosian@kzcr.eu (K.P.); 2Department of Rehabilitation and Sports Medicine, Second Faculty of Medicine, Charles University and University Hospital Motol, 150 06 Prague, Czech Republic; 3Department of Mathematics, Faculty of Science, Jan Evangelista Purkyne University in Usti nad Labem, 400 96 Usti nad Labem, Czech Republic; lucie.loukotova@ujep.cz; 4Department of Orthopaedic Surgery, Faculty of Medicine in Hradec Kralove, Charles University, 500 03 Hradec Kralove, Czech Republic

**Keywords:** percutaneous vertebroplasty, kyphoplasty, complications, compression fractures

## Abstract

**Background**: Percutaneous vertebroplasty (PVP) and kyphoplasty (PKP) are established methods in the treatment of vertebral compression fractures (VCFs). In our manuscript, the target was to evaluate the efficacy of PVPs/PKPs and to determine the implications of potential periprocedural complications. **Methods**: 280 patients, specifically 194 women (69.3%) and 86 men (30.7%), were enrolled. We used the AO spine fractures classification and the Yeom classification to determine the subtype of cement leakage. Only single-level VCFs of the thoracic or lumbar spine were included. Visual analogue scale (VAS) was assessed preoperatively and regularly after the surgery. Vertebral compression ratio (VBCR) was used to determine postoperative vertebral body collapse. **Results**: We recorded 54 cases (19.3%) of cement leakage. There was a significant decrease in mean VAS scores (6.82–0.76 in PVPs, 7.15–0.81 in PKPs). The decrease in VBCR was greater in the VP group (4.39%; 84.21–79.82) compared to the KP group (1.95%; 74.36–72.41). **Conclusions**: No significant difference in the risk of cement leakage when comparing KPs and VPs was found. VPs and KPs provide rapid and significant pain relief in patients with VCFs. Clinically relevant complications of VPs and KPs are rare. Kyphoplasties prevent further vertebral body collapse more effectively compared to vertebroplasties.

## 1. Introduction

Percutaneous vertebroplasty (PVP) and kyphoplasty (PKP) are widely used and respected methods in the therapy of various spinal lesions, especially vertebral compression fractures (VCFs) [1]. The typical indication for a PVP/PKP is an A1 symptomatic VCF, especially in patients with osteoporosis. In these patients, PVPs and PKPs ensure the best therapeutic effect, and they can be crucial in reducing inpatient mortality associated with long–term hospital stays [2]. Furthermore, patients with a history of heart or respiratory diseases, which essentially preclude long bed rest or primary external bracing/casting due to the risk of worsening these conditions, also benefit from early verticalization [3].

The algorithm that is used at our workplace in patients with a thoracic or lumbar VCFs is shown in Figure 1. Most authors recommend performing PVPs or PKPs in fractures that are no more than three weeks old, mainly because of the feasibility of their reduction [2]. On the other hand, Filippiadis et al. (2017) [4] usually requires at least three to four weeks of conservative management prior to surgical intervention. Non-traumatic indications for VP and KP include aggressive vertebral haemangiomas or pathologic osteolytic lesions [2]. On the other hand, PVPs and PKPs should not be performed in patients with neurological symptomatology caused by the fracture. Another contraindication, however, rather relative, are fractures involving the posterior vertebral wall (A3 and A4 types), especially in cases of retropulsion of bony fragments towards the spinal canal [4]. On the contrary, Noguchi et al. (2023) [5] found that the compromise of the posterior vertebral wall does not have to be an absolute contraindication for PVPs and PKPs if we introduce the cement primarily in the ventral third of the vertebral body.

Opinions vary on the efficacy and benefits of prophylactic multi–level vertebral augmentations in osteoporotic patients, but most authors don’t recommend it due to a potentially increased risk of new adjacent vertebral fractures [6]. Other contraindications include bone cement allergy, local or generalized infections and severe bleeding disorders [3].

Like other invasive procedures, PVPs and PKPs are burdened with a particular share of complications with potential clinical sequelae. The most common complication is leakage of the bone cement, which may exhibit various distribution patterns, i.e., prevertebral, epidural, intradiscal, etc. [3]. Other significantly less frequent complications include cement venous embolism and a spectrum of neurological damage (radiculopathy or myelopathy) [7].

In this manuscript, we present a retrospective analysis of 280 patients who underwent a PVP/PKP for a single–level VCF of the thoracic or lumbar spine (134 PKPs and 146 PVPs). All patients underwent surgical procedures at our department within a five-year period (2017–2022). There have been numerous studies discussing the potential graphic complications of PVPs and PKPs, however, we have found little information addressing the long-term effectiveness of these procedures in such a large group. Therefore, we aimed to evaluate the stability of vertebral body height preservation months after the operation, especially considering that all were performed unipedicularly, which usually means a lower amount of applied bone cement. Another crucial topic that we wanted to assess was the rate of complications in our daily practice, in order to compare our results with other authors. We believe that such important insights will be valuable not only for us, but also for other orthopaedic surgeons and spine specialists.

## 2. Materials and Methods

The study was designed as a retrospective, observational and monocentric. It has been conducted in accordance with the ethical standards in the 1964 Declaration of Helsinki and compliance with the approved protocol and Good Clinical Practice. The trial has been reviewed and approved by the Ethics Committee of the Masaryk Hospital in Usti and Labem, Czech Republic. Informed consent to participate in this study and to share medical data was obtained from all patients at the first regular postoperative check-up.

### 2.1. Patient Recruitment

In total, 280 patients have been enrolled in this study. Only patients with single–level traumatic VCFs of the thoracic or lumbar spine verified on preoperative CT scan were included. Table 1 shows the level distribution of the fractures and the determination of their subtypes according to the AO classification [1].

We excluded patients with fractures of two or more adjacent vertebrae, non–traumatic PVPs and PKPs, and cases when vertebral body augmentation was performed as an additional procedure to primary posterior or anterior stabilization. All patients were operated within 72 h after admission to our department. The choice between performing a PVP or PKP was based on the CT scan in conjunction with the algorithm in Figure 1 and partially on individual surgeon’s preference. Typical indications for PVPs and PKPs are shown in Figure 2. To sum up, the most important factor is usually the angle of segmental kyphosis, which is typically associated with the specific subtype of fracture (wedge fractures tend to angulate the vertebral body more [6]). Furthermore, the age and biological state of the patient must be considered (more in discussion and limitations).

### 2.2. Surgical Technique

The procedures were performed under general anaesthesia with the patient positioned prone on an X-ray translucent operating table. We typically initiate the operation with C–arm radiographic verification of the correct spine level in posterior–anterior projection. Subsequently, a Jamshidi needle is inserted into the vertebral body via a transpedicular approach (or slightly extrapedicular in upper thoracic levels to allow sufficient convergence). Then, still, under radiographic control, we start filling the vertebral body with bone cement and check its distribution in lateral projection. In case a KP is performed, a Kirschner Wire (KiWi) is inserted through the lumen of the Jamshidi needle, which is then extracted while the KiWi is kept in place. This step is followed by an approximately one-centimetre-long skin incision for a working cannula. After extracting the KiWi, the cannula serves as a guiding tunnel for a vertebral drill, which creates space for an expandable balloon that reduces the fracture and allows us to form a cavity in the vertebral body for the bone cement. We used two different systems for PVPs, namely the PCD^®^ (Stryker^®^, Kalamazoo, MI, USA) (PCD) and HighV+^®^ (Teknimed, Montredon, France) (HighV) systems. In the PKP group, we used the Spasy™ (SeawonMeditech, Bucheon, Republic of Korea) (Spasy) and the StabiliT^®^ (DFINE, Inc., San Jose, CA, USA) (StabiliT) systems. PCD PVP was used in 25 cases, HighV was used in 121 patients. In the PKP group, Spasy was used in 93 cases and StabiliT in 41 cases. We followed the Yeom classification [2] to determine the subtypes of cement leakage.

### 2.3. Statistical Analysis

To compare the differences in the incidence of these leaks between PVPs and PKPs, being the most prevalent complication, we used the 2-sample test for equality of proportions and the Friedman test to determine the significance of the decrease in postoperative VAS scores. The Friedman test was used due to the ordinal nature of the measured variables and the dependency of individual measurements (VAS scores). All statistic tests were processed through the R software version 4.3.1. The significance level for rejection of the null hypothesis was set at 0.05 for this study, as it provides an acceptable risk of both type I and type II errors.

### 2.4. Follow-Up

After surgery, all patients were transferred back to the standard ward of our department, where they began with postoperative physiotherapy. All patients were fitted with an elastic spine brace or crutches to allow early verticalization and walking. An X-ray (anterior-posterior and lateral projections) was taken on the first postoperative day, where we looked for any signs of cement leakage, i.e., a “graphic complication”. To assess any potential worsening of back pain, we used the Visual Analogue Scale (VAS) regularly during a six-month follow-up period (on the first postoperative day, at six weeks post–op, three months, six months and finally one year after the procedure). We used the three–month post–surgery interval as a decisive period for determining the procedure’s effectiveness according to the patient’s ability to walk independently.

Furthermore, at three months post-op, we always checked if the patients had any need for chronic pain medication (administration once daily or more). Any neurological deficit was assessed according to Frankel’s classification. To determine the potential further vertebral body collapse, we compared the vertebral compression ratio (VBCR) [8] radiographically in both groups (kyphoplasty and vertebroplasty) postoperatively at six weeks and six months after the surgery. The methodology of the measurement is depicted in Figure 3. We reached a general follow–up of one year post-surgery in all patients included in this study.

## 3. Results

Of the 280 enrolled patients, there were 194 women (69.3%) and 86 men (30.7%) in this study with a mean age of 75.5 years (44–90).

### 3.1. Spectrum of Registered Complications

The most frequent complication we recorded was leakage of the bone cement. Yeom C type (leak to a disc adjacent to the intervened vertebra or a leak through a cortical defect) was visible on postoperative radiographs in 14 cases (5%), Yeom S type leak (paravertebral) was found in 13 cases (4.6%). Yeom B type leak (dorsal, epidural) was registered in four cases (1.5%). A cement leakage through the insertion channel in the pedicle of the vertebra was noted in 23 cases (8.2%).

We encountered singular cases of a kyphoplasty balloon rupture and a refracture of the operated vertebra. The above–mentioned radiographic complications are highlighted in Figure 4. We detected no adjacent vertebrae fractures nor any periprocedural neurological complications.

We had a total of 54 cases of cement leakage, of which 29 in the KP group (29/134; 21.64%) and 25 in the VP group (25/146; 17.12%). This difference is statistically insignificant (*p* = 0.34). 85.7% (12 cases) of Yeom C-type leaks were found in the VP group. Examples of different patterns of cement leakages are presented in Figure 5.

### 3.2. VAS Scores and Clinical Outcomes

We recorded a significant decrease in the Visual Analogue Scale (VAS) during the first postoperative year in both groups (KPs and VPs; *p* < 0.001). The mean preoperative VAS was 6.82 in the VP group and 7.15 in the KP group. Six weeks after the surgery, the values dropped to 2.1 and 2.06, respectively, and 0.76 and 0.81 at one-year post-op. The dynamic of the VAS over time is shown in Figure 6.

254 patients (90.7%) were able to walk independently without any restriction or pain, which would require regular administration of analgesics (VAS ≤ 2) at 3 months after the surgery.

### 3.3. Development of VBCR through the Follow-Up

The mean postoperative VBCR in the VP group presented in Figure 7 was 84.21% (67.96–100), 81.18% (47.41–100) at six weeks post-op, and 79.82% (43.17–100) at 6 months post-op. In the KP group, the mean postoperative VBCR was 74.36% (60.26–89.88), 72.48% (57.6–86.62) at six weeks post-op, and 72.41% (57.6–86.62) at six months post-op.

The mean operative time was 10.93 min in the VP group and 18.13 in the KP group.

## 4. Discussion

VCFs in general affect mainly older patients, predominantly women with accelerated postmenopausal or involutional osteoporosis, where even minor trauma affects the spine very often. These VCFs are most indicated for PVP/PKP [1]. We therefore assume, that the demographic structure of our patient group is adequate and in unison with other authors. The purpose of selecting only patients with single-level VCFs was to ensure representative and uniform results in assessing the possibility of fracture reduction. The augmentation of an adjacent vertebral body could potentially lead to alterations in the stiffness of the whole spine segment. These changes could therefore hinder the reduction of the fracture [8]. Furthermore, our department specializes primarily on traumatic VCFs, which in most cases affect fewer vertebrae, as opposed to patients with multiple VCFs due to severe osteoporosis.

One of the technical concerns regarding VPs and KPs is the difference in outcome between unipedicular and bipedicular cement augmentation. Rebolledo et al. (2013) [9] found no significant difference in VAS, restoration of vertebral body height (VBH) or Oswestry Disability Index scores. Lee et al. (2020) [10] came to a similar conclusion of no significant difference in therapeutic outcomes and emphasized the lower amount of cement used in the unipedicular approach. Yan et al. (2014) [11] also mentioned the importance of reducing radiation exposure when implementing the unipedicular approach. We have also experienced conformable clinical results and better cost–effectiveness of unipedicular procedures.

A topic that sparks a debate is the role and duration of preoperative conservative treatment. Son et al. (2014) [12] compared the efficacy of delayed VP versus early VP. The authors concluded that early VP provides prompt pain relief, and fewer cement leaks occur than in DVPs (30% and 59%, respectively). On the other hand, the optimal timing of surgery in patients with VCFs remains a topic that needs to be studied further. Early PVPs or PKPs without an initial trial of conservative therapy may be considered as overtreatment. On the contrary, delayed procedures may not be as effective, not only in reducing a potentially wedged vertebra, but also in dealing with residual pain [13]. We are inclined more towards early VPs because we consider fracture reduction to be sufficiently achievable only in acute fractures. Furthermore, timely verticalization of an elderly patient can play a significant role in their survival in the early post–traumatic period.

There are conflicting opinions on the role of postoperative bracing. Zhang et al. (2019) [14] concluded, that there had been no significant improvement in the outcome of patients who were fitted with a rigid brace for 3 weeks after PVP, compared to a group where no additional bracing was used. On the other hand, Guo et al. (2023) [15] demostrated that postoperative bracing can reduce the risk of refractures of the intervened vertebrae and that it can be effective in reducing residual back pain, which can be crucial for the purpose of early verticalization of elderly patients. Additionally, soft bracing has been proven to be as effective as rigid bracing. We typically fit all our patients with an elastic brace, mainly to prevent further vertebral body sintering and for pain reduction. At the levels of T10 and above, we either leave the patient without external support, or we fit them with crutches for walking. Using soft as opposed to rigid braces might be advantageous in older patients to prevent painful skin ulcers or respiratory compromise [16].

It is generally accepted that cement leakage is common during VPs and KPs, and that it is rarely associated with clinical sequelae. However, ignoring this phenomenon would be erroneous. For example, cement leakage to the space of an intervertebral disc is evidently associated with a higher risk of adjacent vertebra fracture due to the alterations in biomechanical properties of the affected functional spinal unit (FSU) [17]. Moreover, cement leakage into the foraminal or epidural space can lead to various neurological complications, including transient radiculopathy or paraparesis/plegia. Paravertebral leaks are usually asymptomatic, but occasional neuropathy is possible. Leakage into the epidural venous plexus is potentially dangerous due to possible pulmonary cement embolism [17]. One should, therefore, always aim to minimize the risk of potential cement leakage. Zhu et al. (2016) [18] defined some of the factors associated with a higher risk of cement leakage. These include a higher severity level of the fracture (vertebral body collapse > 40%), bipedicular approach or a higher amount of cement applied. We could potentially attribute a fraction of our leakages to the fact, that even some A3 fractures were treated with a PVP or PKP. These cases were always on the borderline, where the patient was not suitable for a larger procedure, due to e.g., severe comorbidities. On the other hand, leaving some of those fracture untreated would most probably lead to severe deformity and persistent pain [5].

The immediate restoration of VBH is probably less important than previously thought. Our group’s differences in postoperative VAS scores are insignificant when comparing VP to KP (*p* > 0.05) in all measured periods. In addition, overcorrection of the angular deformity (segmental kyphosis) can even be linked to a higher risk of adjacent vertebral fractures [19]. On the other hand, restoring VBH and reducing a potential posttraumatic kyphosis may be preferable in the mid–thoracic region to prevent respiratory compromise [18]. Moreover, further collapse and kyphosis of the affected FSU can potentially lead to a higher risk of secondary stenosis of the spinal canal [20]. Our results regarding the VBCR suggest that KPs may provide better linearity in preserving VBH. In other words, the decrease in postoperative VBCR is greater in VPs compared to KPs. However, to ensure the absolute accuracy of the VBCR measurements, we would have had to take regular postoperative CT scans. To sum up, we do not always aim for perfect VBH restoration, but we generally perform a KP in cases where we want to prevent the worsening of potential segmental kyphosis. Another topic related to the restoration of VBH is the choice of device used to reduce the fracure. Numerous devices have been introduced, including expandable balloons (1–4 balloons, depending on the manufacturer), stents or various expandable intravertebral implants that remain in the intervened vertebral body to provide continuous mechanical support. Each implant or device has its own limitations, however, to our knowledge, there has been no study yet that compares all the individual subtypes together. This topic must be studied further in order to distinguish the suitable device/implant for specific patients.

Recently, several authors have started to perform PVPs and PKPs under local anesthesia. For instance, Bonnard et al. (2016) [21] published a series of 95 patients, who received only standardized analgesic medication prior to surgery along with local anesthetic with infiltration of the deep tissues and vertebral periosteum around the planned entry point. They emphasized the advantages of shorter procedural time, lower costs, the opportunity of intraoperative assessment of the patients’ condition or, in some cases, the possibility of bypassing an eventual contraindication for general anesthesia. On the other hand, there were some disadvantages, including discomfort caused by positioning on the operating table or insufficient analgesia. Implementing local anesthesia in indicated patients is one of our long-term goals, considering most of our patients are in their 70s–80s, which is often associated with comorbidities potentially jeopardizing the feasibility of general anesthesia.

Reduced bone mineral density (BMD) is one of the leading factors contributing to the increased risk of new VCFs [22]. In addition, osteopenic and osteoporotic patients are more likely to suffer from adjacent vertebral fractures after PVP/PKP [23]. Apart from BMD, other markers have been introduced to predict the risk of VCFs. For instance, the Vertebral Bone Quality (VBQ) score, which has even been declared to be a more effective predictor of possible future fractures compared to conventional BMD measurements using Dual-X ray Absorptiometry [24]. Additionally, the VBQ excludes possible negative consequences of excessive radiation exposure since it is measured by MRI. We did not routinely measure the BMD, mainly because the patients were typically admitted for a newly developed VCF. Furthermore, the knowledge of BMD would most likely not change our decision-making in these cases.

We consider the share of our graphic and clinical complications to be acceptable in comparison to other authors. For instance, Saracen et al. (2016) [25] studied the largest patient group that we found in the recent literature. They had a total of 863 patients and encountered 410 cases (47.5%) of cement leakage (327 of Yeom S type leaks, 2 Yeom B leaks and 81 Yeom C leaks). The authors recorded 1 asymptomatic pulmonary cement embolization and no neurological complications. Wang et al. (2015) [26] evaluated 107 patients and found 32 cement leaks (Yeom S—10 cases, Yeom C—14 cases, Yeom B—8 cases). No patient had any neurological deficit and again, 1 patient was incidentally diagnosed with asymptomatic pulmonary cement emboli. The graphic complications in our patient group were assessed with conventional radiographs in majority of the cases, therefore, minor cement leaks could have not been detected. However, we don’t consider an asymptomatic cement leak to be an automatic indication for routine postoperative CT scans, mainly because of the risks of cumulative radiation exposure.

### Limitations

This work had some drawbacks, the first being that it was a retrospective study. Furthermore, the individual indications for these procedures will always be slightly biased according to the surgeon’s judgment. Also, more objective measures to track the patients’ status after the surgery (e.g., Oswestry disability index) would be useful. This factor was notable especially when evaluating the clinical significance of the cement leaks. The assessment of a potential neurological deficit as a result of e.g., epidural leakage is quite clear. However, it was not possible to attribute potential indifferent postoperative back pain to a specific cause. Especially in intradiscal or paravertebral leakages, we did not find any peculiarities in the development of the patient’s condition in comparison to patients with excellent radiographic findings. This aspect should be assessed in future studies. Finally, although we attempted to objectify the patient recruitment, the surgeon preferences caused a certain case-by-case bias in choosing between PVP and PKP.

## 5. Conclusions

PVPs and PKPs represent advantageous treatment modalities for A1 and some A3 VCFs. Clinically relevant complications are rare. Preferring KPs in A1 VCFs with >10° segmental kyphosis may lead to better radiographic results. Better VBCR does not equate to better clinical outcome. There are several areas suitable for further research. This includes the optimal timing of the procedures, comparing different implants and reduction devices, or designing new outputs to synchronize radiographic findings and clinical outcomes more effectively.

## Figures and Tables

**Figure 1 jcm-13-01495-f001:**
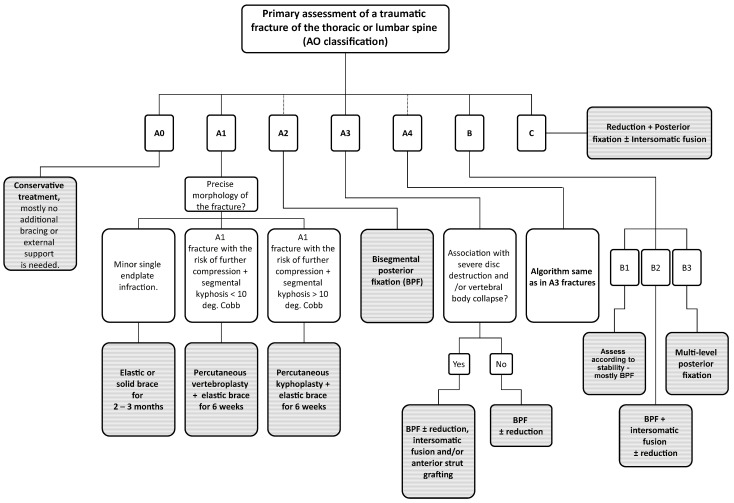
Visualization of the therapeutical algorithm for patients with thoracic or lumbar spine fractures. The grey boxes indicate our preferred solutions for individual types of fractures. In fractures A2-A4, B & C, consider additional bracing after surgery according to fracture stability and morphology. Jewett limited–contact orthosis for highly unstable thoracolumbar junction fractures (roughly Th11-L2), TLSO orthosis in segments below L2, and Chêneau brace for levels above Th11. In cases of high primary stability post-surgery, an elastic brace is sufficient. If the patient’s body composition is abnormal (extreme obesity) or unable to put on the brace independently, consider using crutches for verticalization and walking. In fractures A3–A4, B & C, always perform spinal decompression in case of preoperative neurologic deficit.

**Figure 2 jcm-13-01495-f002:**
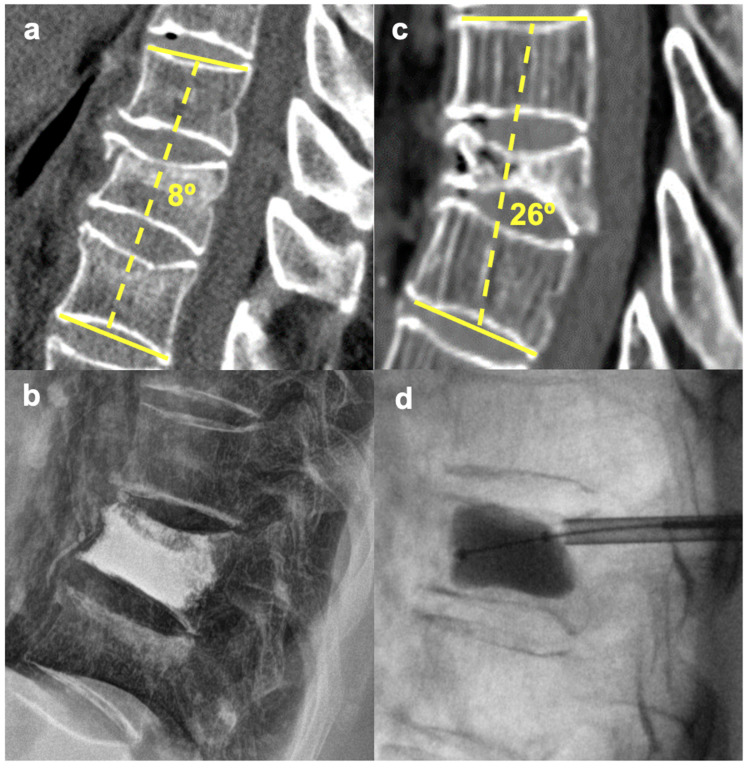
Difference in the indications for PVP vs. PKP. (**a**,**b**)—PVP for an A1 L1 fracture with 8° segmental kyphosis. (**c**,**d**)—PKP (balloon expansion with reduction) for an A1 T9 fracture with 26° segmental kyphosis.

**Figure 3 jcm-13-01495-f003:**
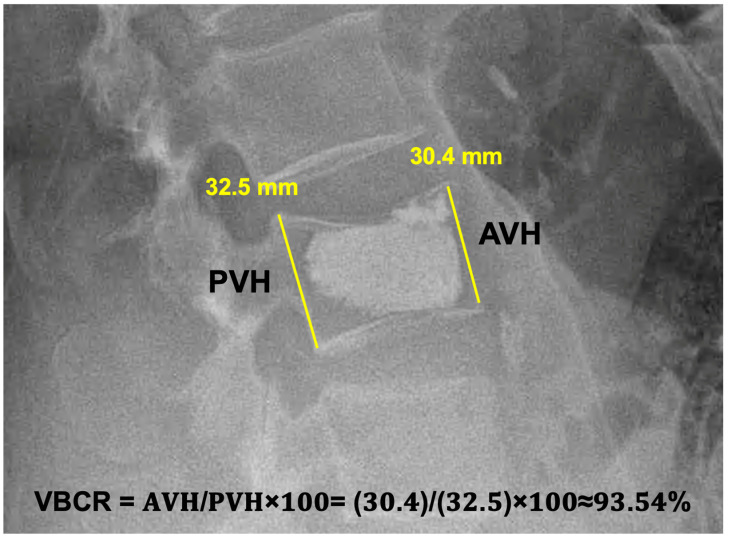
A postoperative lateral X-ray of an L3 vertebroplasty measuring the Vertebral Compression Ratio. AVH = Anterior Vertebral Height; PVH = Posterior Vertebral height. The AVH and PVH measurement lines should be in contact with the vertebral body’s anterior and posterior margins.

**Figure 4 jcm-13-01495-f004:**
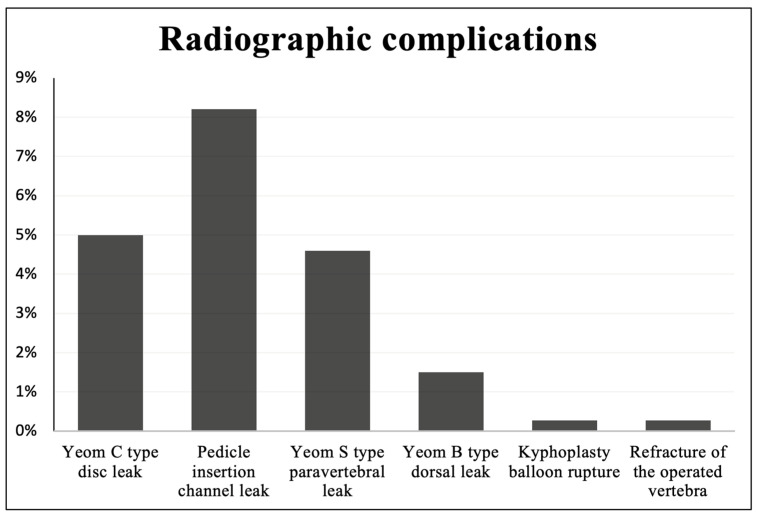
Radiographic complications encountered during percutaneous vertebroplasty and kyphoplasty.

**Figure 5 jcm-13-01495-f005:**
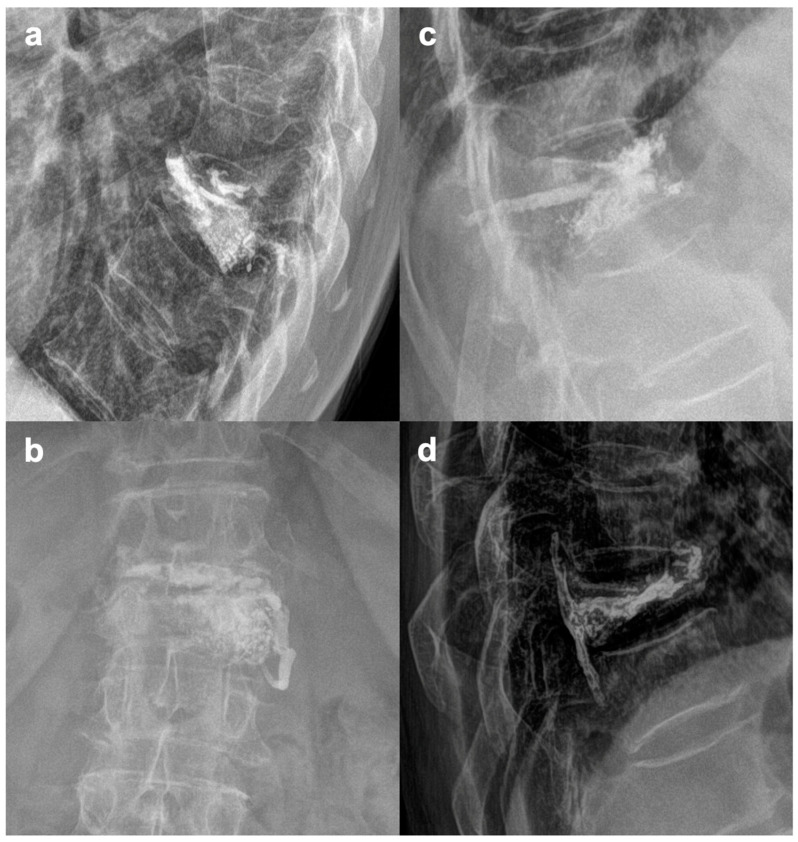
Various patterns of cement leakages. (**a**)—Yeom C (intradiscal); (**b**)—Yeom S (paravertebral), (**c**)—Pedicle insertion channel leak; (**d**)—Yeom B (dorsal, epidural).

**Figure 6 jcm-13-01495-f006:**
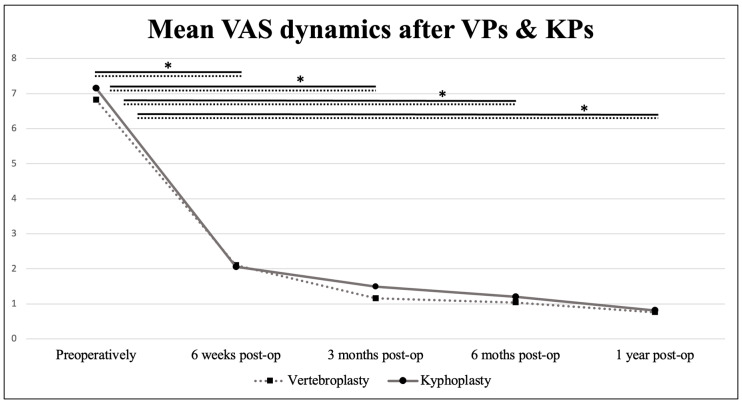
Mean Visual analogue scale dynamics in the postoperative period. Values connected by lines differ significantly from each other (* *p* < 0.005).

**Figure 7 jcm-13-01495-f007:**
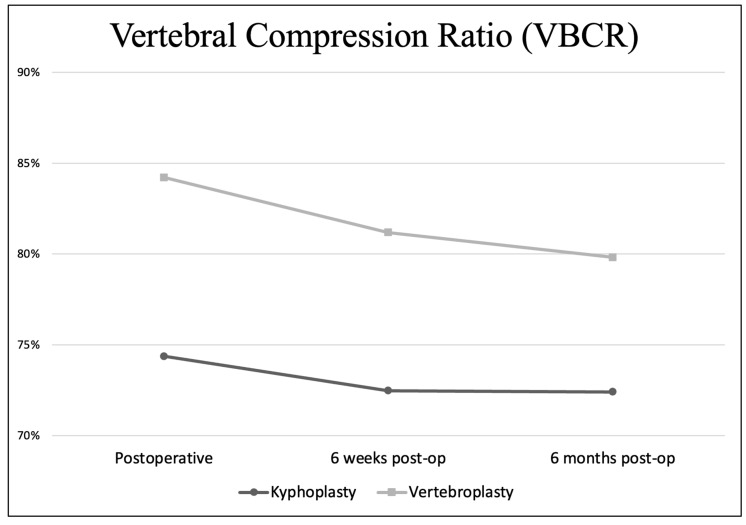
Vertebral compression ratio in the postoperative period.

**Table 1 jcm-13-01495-t001:** Level distribution of the operated fractures and the types of operated fractures (according to the AO classification).

Level Distribution of the Operated Fractures		Type of Fracture (AO Classification)	
Th8 Th9 Th10 Th11 Th12 L1 L2 L3 L4	3 cases (1%) 1 case (0.4%) 4 cases (2%) 20 cases (7%) 71 cases (25%) 100 cases (36%) 47 cases (17%) 25 cases (9%) 9 cases (3%)	A1	228 cases (82%)
		A3	50 cases (18%)

## Data Availability

Data are contained within the article.

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
