# Peer review of "Efficacy and Complication Rates of Percutaneous Vertebroplasty and Kyphoplasty in the Treatment of Vertebral Compression Fractures: A Retrospective Analysis of 280 Patients"

_jcm, 2024, doi:10.3390/jcm13051495_

Round 1
Reviewer 1 Report
Comments and Suggestions for Authors
The prognosis may vary depending on bone density levels. It is believed that study on patients' bone mineral density levels is needed.
Is there are no cases in which fractures of adjacent segments occurred after surgery?
I wish the above contents were added.
Author Response
Reviewer 1
Firstly, on behalf of all the co-authors, we would like to thank the reviewer for taking their valuable time to elaborate on our manuscript. The comments of the reviewer were very apt and allowed us to optimize the structure of our work. We tried our best to meet the requirements in individual points and sincerely hope, that our manuscript is acceptable in its present form for publication in your highly esteemed journal. All revisions are indicated in red font in text (visible corrections form).
- The prognosis may vary depending on bone density levels. It is believed that study on patient’s bone mineral density level is needed.
Thank you for this valuable suggestion, we added a paragraph in the “Discussion” section (page 10, line 343-353) dealing with bone mineral density. Unfortunately, since the study was retrospective and some of the patients have already been dismissed from regular follow-up, we were not able to provide a sufficient study of the patients’ bone mineral density in our manuscript.
- Are there no cases in which fractures of adjacent segments occured after surgery?
Thank you for this comment. We encountered no such complication – we added this comment in the “Results” section (page 6, subsection “Spectrum of registered complications”, line 201).
Reviewer 2 Report
Comments and Suggestions for Authors
Title: Why the phrase 'still relevant and safe?"
If safety and relevance were specific concerns, the reasons must be elaborated in intro
Pls modify title
Abstract: Needs to be substantially modified
The demographic resuts have been presented in methodology section. The methodology section is very unclear
Very long conclusion
Intro: The major part of the intro seems to focus on phrases such as "we follw", "we approach", we usuall;y perform" etc. The intro must be more generalised
The relevant lacuna in the available literature and the question needing to be answered may be presented
The current form is such that the authors already have a strong bias towards to the conclusions
Methods: Reasonably well explained
Grammatical and typographical errors may be corrected
Results: May be presented under specific subheadings
Good images (including surgical illustrations) must be added
Discussion: Very short
The purpose of study must be clearly explained
Relevant literature must be discussed
May be presenetd under pecific subheadings
Pls add the limitations and conclusions clearly
The personalised pharases need to be substantially modified
Comments on the Quality of English Language
Overall reasonably written. Some corrections are necessary
Author Response
Reviewer 2
Firstly, on behalf of all the co-authors, we would like to thank the reviewer for taking their valuable time to elaborate on our manuscript. The comments of the reviewer were very apt and allowed us to optimize the structure of our work. We tried our best to meet the requirements in individual points and sincerely hope, that our manuscript is acceptable in its present form for publication in your highly esteemed journal. All revisions are indicated in red font in text (visible corrections form).
- Modify the title – why the phrase “still relevant and safe?”
Thank you for this comment. We modified the title (page 1; lines 2-4) accordingly.
- Modify the abstract.
Thank you for this comment. We modified the Abstract according to the JCM guidelines (page 1, lines 18-33).
- The demographic results have been presented in methodology section. The methodology section is very unclear.
Thank you for this comment. We went through the “Materials and Methods” section, divided it into specific subheadings and tried to differentiate the information properly.
- Very long conclusion.
Thank you for this comment. We shortened the “Conclusions” section and tried to make it more apt.
- Intro: The major part of the intro seems to focus on phrases such as “we follow”, “we approach”, “we usually perform” etc. The intro must be more generalized.
Thank you for this comment. We removed the personal phrases and modified the “Introduciton” section (page 2, lines 46-49).
- The relevant lacuna in the available literature and the question needing to be answered may be presented.
Thank you for this comment. We added this information in the “Introduction” section (page 2, lines 73-81).
- The current form is such that the authors already have a strong bias towards to the conclusions.
Thank you for this comment. We tried to modify the manuscript to make it less one-sided with elaborating more on the discussion, mentioning limitations and rephrasing the conclusion.
- Grammatical and typographical errors may be corrected.
Thank you for this comment. We went through the manuscript and tried to correct grammatical errors.
- Results: May be presented under specific subheadings.
Thank you for this comment. We modified the “Results” section and divided it into specific subheadings as requested (pages 5-10).
- Good images (including surgical illustrations) must be added.
Thank you for this comment. We added 2 new Figures, including intraoperative images and radiographic complications (Figures 2 & 5, pages 5 & 8).
- Discussion: Very short.
Thank you for this comment. We elaborated thoroughly on the “Discussion” section, including paragraphs dealing with the role of bracing, risk factors for cement leakage, local anesthesia, bone mineral density and more (pages 8-11).
- The purpose of study must be clearly explained.
Thank you for this comment. We tried to explain the main purposes of the study in the “Introduction” section (page 2, lines 73-81).
- Relevant literature must be discussed.
Thank you for this comment. We added 10 more references and implemented them in the “Discussion” section.
- May be presented under pecific subheadings.
Thank you for this comment. We modified the sections “Materials and Methods” and “Results” and divided them into specific subheadings as requested.
- Pls add the limitations and conclusions clearly.
Thank you for this comment. We added a specific subsection “Limitations” in the “Discussion” section (pages 12-13, lines 373-381). The whole “Conclusions” section has been rewritten (page 13, lines 383-389).
- The personalised phrases need to be substantially modified.
Thank you for this comment. We removed the personal phrases and modified the “Introduciton” section (page 2, lines 46-49). Furthermore, we went through the phrases once again and tried to make it more evidence based.
Reviewer 3 Report
Comments and Suggestions for Authors
- Thanks for the manuscript entitled "Percutaneous Vertebroplasty and Kyphoplasty in the Treatment of Vertebral Compression Fractures – Relevance and Safety: A Retrospective Analysis of 280 Patients". There are several comments regarding this manuscript.
-
Clarity of Introduction: The introduction provides a comprehensive overview of percutaneous vertebroplasty (VP) and kyphoplasty (KP) as treatment methods for vertebral compression fractures (VCFs). The significance and typical indications for these procedures are well-explained. However, it would be beneficial to explicitly state the research question or hypothesis guiding the study.
-
Study Design and Population: The study design, a retrospective analysis, is clearly stated. The inclusion and exclusion criteria are well-defined, but it would be helpful to elaborate on the rationale behind the specific criteria, such as the decision to include only single-level traumatic VCFs.
-
Ethical Considerations: Ethical standards and the approval process are briefly mentioned. Please provide more details on how patient consent and data confidentiality were managed in a retrospective study, considering the data spans five years.
-
Patient Demographics: The demographic information is adequately presented. However, it would be valuable to discuss whether the demographic characteristics of the sample are representative of the broader population with VCFs.
-
Treatment Procedures: The surgical procedures are described clearly, but it would be beneficial to provide more insight into the decision-making process regarding the choice between VP and KP, especially considering the surgeon's preferences.
-
Complications and Clinical Outcomes: The complications, especially cement leakage, are well-documented. The clinical outcomes, as measured by the Visual Analog Scale (VAS), are appropriately presented. However, please elaborate on any limitations or challenges in assessing the clinical significance of cement leakage, given its potential asymptomatic nature.
-
Statistical Analysis: The statistical methods are appropriately mentioned, including the statistical tests used. However, consider providing more details on the rationale behind choosing specific tests and the significance level.
-
Results Interpretation: The interpretation of results, especially regarding the lack of statistical significance in cement leakage between KP and VP, is clear. However, discuss potential clinical implications and the impact on the choice between these procedures.
-
Discussion and Limitations: The discussion section provides a thorough analysis of the study outcomes. However, consider discussing the broader clinical implications, the generalizability of findings, and potential avenues for future research. Additionally, acknowledge the limitations of the retrospective design and any potential biases introduced.
-
Conclusion: The conclusion is concise and summarizes key findings. Reiterate the main implications for clinical practice and potential areas for future research.
-
Overall Impression: The study offers valuable insights into the safety and efficacy of VP and KP for VCFs. Strengthening certain sections, addressing specific points in the reviewer's comments, and providing additional context will enhance the manuscript's overall quality.
Please revise the manuscript accordingly and provide detailed responses to each comment for further consideration.
Author Response
Reviewer 3
Firstly, on behalf of all the co-authors, we would like to thank the reviewer for taking their valuable time to elaborate on our manuscript. The comments of the reviewer were very apt and allowed us to optimize the structure of our work. We tried our best to meet the requirements in individual points and sincerely hope, that our manuscript is acceptable in its present form for publication in your highly esteemed journal. All revisions are indicated in red font in text (visible corrections form).
- Clarity of introduction: “..it would be beneficial to explicitly state the research question or hypothesis guiding the study.”
Thank you for this comment. We tried to explain the main purposes of the study in the “Introduction” section (page 2, lines 73-81).
- Study design and population: “..it would be helpful to elaborate on the rationale behind specific criteria, such as the decision to include only single-level traumatic VCFs.”
Thank you for this comment. We explained our rationale behind the selection criteria in the “Discussion” section (page 10, lines 253-259).
- Ethical considerations: “..provide more details on how patient consent and data confidentiality were managed..”
Thank you for this comment. We added this comment in the “Materials and Methods” section (page 2, lines 87-88).
- Patient Demographics: “..it would be valuable to discuss whether the demographic characteristics of the sample are representative of the broader population with VCFs.”
Thank you for this comment. We elaborated on this topic in the “Discussion” section (page 10, lines 249-252).
- Treatment procedures: “..it would be beneficial to provide more insight into decision-making process regarding the choice between VP and KP..”
Thank you for this comment. We described our decision-making process more thoroughly in the “Materials and Methods” section (page 3, lines 103-109).
- Complications and Clinical Outcomes: “..elaborate on any limitations or challenges in assessing the clinical significance of cement leakage..”
Thank you for this comment. We elaborated on the limitations in the “Discussion” section (pages 12-13 – subsection “Limitations”, lines 373-381).
- Statistical Analysis: “..consider providing more details on the rationale behind choosing the specific tests and the significance level.”
Thank you for this comment. We provided some more details in the Statistical analysis (pages 3-4, subsection “Statistical Analysis”, lines 138-142).
- Results Interpretation: “..discuss potential clinical implications and the impact on the choice between these procedures.”
Thank you for this comment. We added comments about possible implications of choosing these procedures in specific scenarios in the “Discussion” section (page 11, lines 304-309).
- Discussion and Limitations: “..consider discussing the broader clinical implications, the generalizability of findings, and potential avenues for future research.”
Thank you for this comment. We added several insights and tried to elaborate on this topic in the “Discussion” section (pages 10-11-12, lines 272-276 and 323-342).
- Conclusion: “Reiterate the main implications for clinical practice and potential areas for future research.”
Thank you for this comment. We tried to rephrase the “Coclusion” section accordingly (page 13, lines 383-389).
Round 2
Reviewer 1 Report
Comments and Suggestions for Authors It has been well edited and is suitable for publication.Reviewer 2 Report
Comments and Suggestions for Authors
The recommended changes may be added
The manuscript may be accepted in the current form
Comments on the Quality of English LanguageOverall well written
Some changes are necessary